# Learn to Match with No Regret: Reinforcement Learning in Markov Matching Markets

**Yifei Min**
Department of Statistics and Data Science
Yale University
New Haven, CT 06511
yifei.min@yale.edu

**Tianhao Wang**
Department of Statistics and Data Science
Yale University
New Haven, CT 06511
tianhao.wang@yale.edu

**Ruitu Xu**
Department of Statistics and Data Science
Yale University
New Haven, CT 06511
ruitu.xu@yale.edu

**Zhaoran Wang**
Departments of Industrial Engineering
and Management Sciences
Northwestern University
Evanston, IL 60208
zhaoranwang@gmail.com

**Michael I. Jordan**
Department of Electrical Engineering
and Computer Science
University of California, Berkeley
CA 94720
jordan@cs.berkeley.edu

**Zhuoran Yang**
Department of Statistics and Data Science
Yale University
New Haven, CT 06511
zhuoran.yang@yale.edu

## Abstract

We study a Markov matching market involving a planner and a set of strategic agents on the two sides of the market. At each step, the agents are presented with a dynamical context, where the contexts determine the utilities. The planner controls the transition of the contexts to maximize the cumulative social welfare, while the agents aim to find a myopic stable matching at each step. Such a setting captures a range of applications including ridesharing platforms. We formalize the problem by proposing a reinforcement learning framework that integrates optimistic value iteration with maximum weight matching. The proposed algorithm addresses the coupled challenges of sequential exploration, matching stability, and function approximation. We prove that the algorithm achieves sublinear regret.

## 1 Introduction

Large-scale digital markets play a crucial role in modern economies, and yet the analysis and design of such ever-changing markets demand powerful instruments beyond economics. An in-depth understanding of the dynamically changing market environments requires our methods to be adaptive, scalable, and incentive compatible in the face of significant nonstationarity. In particular, massive data streams arising from digital markets provide us with a great opportunity to meet such challenges through learning-based mechanism design. A recent line of work applies modern machine learning algorithms to classic problems in adaptive mechanism design (Jagadeesan et al., 2021; Liu et al., 2021; Sankararaman et al., 2021; Basu et al., 2021; Liu et al., 2022), and among which one especially

36th Conference on Neural Information Processing Systems (NeurIPS 2022).

consequential aspect of learning-aware market design is that of a matching market, a course of problems central to microeconomics (Mas-Colell et al., 1995). Existing work focuses primarily on static matching markets. However, the more challenging yet critically important setting of dynamic matching markets has been neglected. In this paper, we are going to provide a reinforcement learning (RL) based solution that enables a data-driven treatment to such dynamic matching market problems.

To begin with, we propose a *Markov matching market* model with a central planner and a set of two-sided agents. The market is defined with a time horizon $H \in \mathbb{Z}_+$, and a set of agents $I_h \cup J_h$ enter the market in context $C_h$ at step $h \in [H]$ with unknown utility functions $u_h$ and $v_h$ for agents in $I_h$ and $J_h$, respectively. We assume that the agents' utilities $u_h$ and $v_h$ depend on the context $C_h$. Given the utility functions, the agents from both sides seek to achieve a myopic stable matching with transferable utilities at each step (Shapley and Shubik, 1971). In particular, the contexts $C_h$ are subject to a Markov transition kernel controlled by the planner's policy, and the goal of the planner is to select an optimal policy on the contexts, which together with the stable matching among the agents maximizes the expected accumulated social welfare.

As an illustration, we may consider the model as a simplified abstraction of a ride-hailing platform, where the horizon $H$ is set as the time span of a day (Qin et al., 2020; Özkan and Ward, 2020; Hu and Zhou, 2021). Under such a scenario, the ride-hailing platform acts as the central planner while the two-sided agents $I_h$ and $J_h$ are the drivers and the riders. The contexts $C_h$ at each time step may retain an assortment of ride-related metrics such as GPS location, car types, and pricing. The platform is incentivized to ensure the stability of the matchings between the riders and the drivers at each step so that they do not prefer alternative matching outcomes. They may leave the platform otherwise. Moreover, the platform also aims to provide a policy that maximizes the accumulated social welfare as a measure of the level of satisfaction for both parties.

As illustrated in the ride-hailing platform example, there are several key challenges to achieving an effective Markov matching market. First, the agents' utilities and the transition of contexts are unknown, and we need to perform efficient explorations to collect the required information on the utilities and the transitions. Second, we need to guarantee the stability of the matching in each step, and a naive consideration of only the difference in total utilities is insufficient, as discussed in Jagadeesan et al. (2021). To this end, we adopt a notion of *Subset Instability* (SI) from Jagadeesan et al. (2021) as a metric that quantifies the distance between a proposed matching and the optimal stable matching. This metric has the flavor of Shapley value, comparing the discrepancy between the total utilities over all subsets of the participating agents while accounting for the utility transfers. For an efficient estimation, it is also important to take into account the function class of the utilities as well as the Markov transition kernel, thus demanding a systematic usage of function approximation.

To tackle these challenges, we develop a novel algorithm called *Sequential Optimistic Matching* (SOM), which features a combination of optimistic value iteration and max-weight matching. Note that the planner's problem can be treated as a standard Markov decision process (MDP) if we regard the value of the max-weight matching over true utilities as its reward. Inspired by this observation, on the agents side, our algorithm applies the optimism principle to construct UCB estimates of the utilities. The algorithm computes the corresponding max-weight matching based on these UCB estimates, and the values of the resulting matching serve as the surrogate rewards. Although we do not have access to an unbiased estimate of the rewards, the key property here is that the surrogate rewards upper bound the true social welfare, thus justifying optimistic planning by the planner. Interestingly, our framework can readily incorporate any online RL algorithm based on the principle of optimism.

We show that the suboptimality of the accumulated social welfare for our algorithm consists of two parts: (1) the planner's regret in terms of the suboptimality of the policy, and (2) the agents' regret in terms of SI. Jagadeesan et al. (2021) proved that SI can be bounded by the sum of the optimistic bonuses, and we further show that the planner's regret can also be bounded by the bonus sum. In this way, we reconcile the seemingly independent learning goals of the planner and the agents, and thereby provide a unified approach to controlling the suboptimality of the total social welfare. In particular, based on the above decomposition, we further show that in the case of linear function approximation, our algorithm enjoys a sublinear regret independent of the size of the context space.

Our results provide a uniform treatment of dynamic matching markets via online RL. In particular, our framework incorporates a complete matching problem in each time step compared with existing methods on online RL such as `LSVI-UCB` (Jin et al., 2020), and our model provides an extension to a dynamic setup with transitions between contexts, which is also beyond the existing setups of

matching bandits (Jagadeesan et al., 2021). More importantly, our work lies beyond a straightforward combination of online RL with matching bandits due to the unique technical challenges to be discussed in Section 4 and further elaborated in Section C.

Our main contributions are summarized as follows:

(i) We propose a novel Markov matching market model that captures a range of instances of centralized matching problems.

(ii) We develop a novel algorithm that combines optimistic value iteration with max-weight matching, such that any online RL algorithm based on optimism can be readily incorporated into the framework.

(iii) We provide a general analysis framework and show that our proposed algorithms achieve sublinear regret under proper structural assumptions on the underlying model.

## 2  Related Work

There is an emerging line of research on learning stable matchings with bandit feedback (Das and Kamenica, 2005; Liu et al., 2020, 2021; Sankararaman et al., 2021; Cen and Shah, 2021; Basu et al., 2021) using the mature tools from the bandit literature. Most of them focus on matchings with non-transferable utilities (Gale and Shapley, 1962), which fails to capture real-world markets with monetary transfers between agents, e.g., payments from passengers to drivers on ride-hailing platforms. The study of learning for matchings with transferable utilities is comparably limited, and our work extends Jagadeesan et al. (2021) to dynamic scenarios in this regime. Broadly speaking, our work is also related to the research on learning economic models via RL. In particular, Kandasamy et al. (2020); Rasouli and Jordan (2021) studied VCG mechanisms, and Guo et al. (2021) studied exchange economies. Although similar in spirit, these models differ from our Markov matching markets in their mathematical structure, and thus have different solution concepts and planning methods.

Our model of Markov matching markets is related to the topic of dynamic matching in the economics literature (Taylor, 1995; Satterthwaite and Shneyerov, 2007; Niederle and Yariv, 2009; Ünver, 2010; Anderson et al., 2014; Lauermann and Nöldeke, 2014; Leshno, 2019; Akbarpour et al., 2020; Baccara et al., 2020; Loertscher et al., 2018; Doval and Szentes, 2019). Instead of studying the learning problem in matching markets, the goal therein is mainly focused on the problem of optimal mechanism design (Akbarpour et al., 2014) with known utilities or explicit modelling of agents' arrivals via queuing models (Zenios, 1999; Gurvich and Ward, 2015). In this work we focus on the notion of (static) stability of myopic matchings (Shapley and Shubik, 1971). There is also a line of literature on the notion of dynamic stability for matching markets (Damiano and Lam, 2005; Doval, 2014; Kadam and Kotowski, 2018; Doval, 2019; Kotowski, 2019; Kurino, 2020; Liu, 2020), and it is an interesting open problem to study learning for dynamically stable matchings.

Our methodology builds upon recent progress in online RL, where the "optimism in face of uncertainty" principle has engendered efficient algorithms that are either model-based (Jaksch et al., 2010; Osband et al., 2016; Azar et al., 2017; Dann et al., 2017) or model-free (Strehl et al., 2006; Jin et al., 2018; Fei and Xu, 2022; Fei et al., 2021a), and can be combined with function approximation techniques (Yang and Wang, 2019; Jin et al., 2020; Zanette et al., 2020; Ayoub et al., 2020; Wang et al., 2020; Fei et al., 2021b; Yang et al., 2020; Zhou et al., 2021; Min et al., 2021a,b; Du et al., 2021; Jin et al., 2021). We note that these approaches can be incorporated into our framework with proper modifications on structural assumptions and correspondingly the algorithm. We do not pursue these extensions here, however, as our regret analysis is already sufficiently challenging given the lack of an unbiased estimate of the reward, and the additional constraints imposed by the requirements of matching stability.

## 3  Preliminaries

### 3.1  Markov Matching Markets

We first review basic concepts for matching with transferable utilities (Shapley and Shubik, 1971). Consider a two-sided matching market where $\mathcal{I}$ denotes the set of all side-1 agents (e.g., buyers)

and $\mathcal{J}$ denotes the set of all side-2 agents (e.g., sellers). Given any set of participating agents $I \times J \in 2^{\mathcal{I}} \times 2^{\mathcal{J}}$, a matching $X$ is a set of pairs $(i, j)$ indicating $i \in I$ is matched with $j \in J$, and each agent can be matched at most once. For any pair of agents $(i, j) \in I \times J$, we denote by $u(i, j)$ the utility of agent $i$ and $v(i, j)$ the utility of agent $j$ when they are matched. In addition to the matching $X$, we also allow transfers between agents, summarized by the transfer function $\tau : I \cup J \to \mathbb{R}$. For each agent $i \in I \cup J$, $\tau(i)$ is the transfer that it receives. We assume that the transfers are within agents, so $\sum_{i \in I \cup J} \tau(i) = 0$.

The overall market outcome is denoted by a tuple $(X, \tau)$, where $X$ represents the matching and $\tau$ represents transfers. For any $(i, j) \in X$, the total utilities for agent $i$ and $j$ are $u(i, j) + \tau(i)$ and $v(i, j) + \tau(j)$ respectively. Moreover, if no agents prefer any alternate outcome, then we say $(X, \tau)$ is stable (see Definition B.1 for details). The stable matching can be found by solving the corresponding max-weight matching, as will be explained in Section 4.1.

Based on these (classical) definitions, we formulate the notion of a *Markov matching market* involving a planner and a set of two-sided agents. Throughout the paper, we focus on matchings with transferable utilities between two-sided agents, and we may omit such descriptions for convenience.

**Definition 3.1** (Markov matching markets). A Markov matching market is denoted by a tuple $M = (\mathcal{C}, \Upsilon, \{I_h\}_{h=1}^H, \{J_h\}_{h=1}^H, \{\mathbb{P}_h\}_{h=1}^H, \{u_h\}_{h=1}^H, \{v_h\}_{h=1}^H)$. Here $\mathcal{C}$ is the set of contexts and $\Upsilon$ denotes the set of planner's actions. At each step $h \in [H]$, $I_h \cup J_h \subset \mathcal{I} \times \mathcal{J}$ is the set of participating agents, and $u_h : \mathcal{C} \times \Upsilon \times \mathcal{I} \times \mathcal{J} \to \mathbb{R}$ and $v_h : \mathcal{C} \times \Upsilon \times \mathcal{I} \times \mathcal{J} \to \mathbb{R}$ are utility functions for two sides of agents respectively. For each $h \in [H]$, $\mathbb{P}_h(C' \mid C, e)$ is the transition probability for context $C$ to transit to $C'$ given action $e$.

In such Markov matching markets, the learning goal is two-fold: (1) to learn the stable matching in each step and (2) to maximize the accumulated social welfare.

## 3.2 A Reinforcement Learning Approach to Markov Matching Markets

We now translate the problem into the language of RL. Given Definition 3.1, we consider an episodic setting with $K$ episodes where each episode consists of $H$ steps of sequential matchings. Each episode proceeds in the following way: at each step $h \in [H]$ under context $C_h$, a set of agents $I_h \cup J_h$ enter the market. The planner takes action $e_h$, implements the matching $(X_h, \tau_h)$, and observes the noisy feedback of utilities $u_h(C_h, e_h, i, j)$ and $v_h(C_h, e_h, i, j)$ for all $(i, j) \in X_h$. Then the context transits according to $\mathbb{P}_h(\cdot \mid C_h, e_h)$, and the market proceeds to the next step.

Note that here the implemented matching in each step is myopic and we seek for stability for each of these matchings. The maximization of the accumulated social welfare is achieved through the planner's actions $\{e_h\}_{h=1}^H$ that control the transitions of contexts, which together with planner's actions determine the optimal values of the matchings. To apply RL techniques to maximize the accumulated social welfare, let us specify the ingredients of the corresponding RL problem.

**States and Actions.** The state space is $\mathcal{S} = \mathcal{C} \times 2^{\mathcal{I}} \times 2^{\mathcal{J}}$. The action space is $\mathcal{A} = \Upsilon \times \mathcal{X} \times \mathcal{T}$ where $\mathcal{X}$ denotes the set of all matchings and $\mathcal{T}$ is the set of all possible transfers among the agents. For each step $h \in [H]$, the state $s_h = (C_h, I_h, J_h)$ contains the context and participating agents, and the action $a_h = (e_h, X_h, \tau_h)$ contains the planner's action $e_h$ and the matching outcome $(X_h, \tau_h)$ for the agents.

**Rewards.** At each step $h \in [H]$, given the state $s_h = (C_h, I_h, J_h)$ and the action $a_h = (e_h, X_h, \tau_h)$, the immediate reward is the social welfare (i.e. sum of utilities):

$$r_h(s_h, a_h) := \sum_{(i,j) \in X_h} [u_h(C_h, e_h, i, j) + v_h(C_h, e_h, i, j)]. \tag{1}$$

Note that the transfer $\tau_h \in \mathcal{T}$ does not appear in the reward since the total transfer sums to zero.

**Transition of States.** The state consists of the context in $\mathcal{C}$ and the sets of agents in $2^{\mathcal{I}} \times 2^{\mathcal{J}}$. The transition of context at step $h$ follows the heterogeneous transition function $\mathbb{P}_h(C_{h+1} \mid C_h, e_h)$, which only depends on the planner's action $e_h$ and is independent of the matching $(X_h, \tau_h)$.

We assume that the sequence of agent sets $\{I_h, J_h\}_{h=1}^H$ is generated independently from other components in this matching market. We also assume the same sequence through all $K$ episodes for the sake of clarity. Note that $I_h$ and $J_h$ can also be handled as part of the context $C_h$ and covered by

our current argument with more involved transition dynamics, which is often task-specific. Our minor simplification serves to build a generic framework and avoid detailed modeling of the agent sets.

**Policies and Value Functions.** A policy $\pi$ is defined as $\pi = \{\pi_h\}_{h=1}^{H}$, where for each $s \in \mathcal{S}$, $\pi_h(\cdot|s)$ is a distribution on $\mathcal{A}$. The policy consists of two parts: the planner's part (i.e., choosing $e \in \Upsilon$ to influence the market context $C$), and the agents' part (i.e., determining the matching-transfer $(X, \tau)$). We use $\Pi$ to denote the set of all such policies.

For any policy $\pi$, we define each value function $V_h^{\pi}(\cdot)$ as

$$V_h^{\pi}(s) := \mathbb{E}_{\pi}\left[\sum_{l=h}^{H} r_l(s_l, a_l) \;\middle|\; s_h = s;\; a_l \sim \pi_l(\cdot|s_l),\; s_{l+1} \sim \mathbb{P}_l(\cdot|s_l, a_l), \forall\, h \leq l \leq H\right]. \quad (2)$$

Maximizing the accumulated social welfare is equivalent to maximizing $V_1^{\pi}$ over $\pi \in \Pi$, so the overall performance over $K$ episodes is evaluated through the regret

$$R(K) := \sum_{k=1}^{K}\left[\max_{\pi \in \Pi} V_1^{\pi}(s_1) - V_1^{\pi_k}(s_1)\right], \quad (3)$$

where $\pi_k$ denotes the policy in episode $k$. See Appendix A for detailed definitions of our notation.

# 4 Method: An Optimistic Meta Algorithm

The dynamic matching problem introduced in Section 3 faces several coupled challenges. Specifically: 1) we study the dynamic setting and therefore the algorithm proposed by (Jagadeesan et al., 2021) for the bandit setting (i.e., $H = 1$) does not apply; 2) the dynamic regime also requires us to handle the propagation of error through $H$ steps in the analysis; 3) compared with standard MDPs in RL, here we aim to maximize the social welfare and thus the regret involves a new notion called Subset Instability, which is a nonlinear functional of the rewards of the agents. Regret decomposition involving such a complex quantity was never considered in previous works (Jaksch et al., 2010; Osband et al., 2016; Azar et al., 2017; Jin et al., 2020; Ayoub et al., 2020; Wang et al., 2020; Fei et al., 2021b; Yang et al., 2020; Zhou et al., 2021).

We now introduce an algorithm for the Markov matching market that addresses all these challenges. This algorithm involves solving a sequence of combinatorial optimization problems to obtain a UCB estimate (Section 4.1). Moreover, for the purpose of the theoretical analysis, as the regret is in terms of Subset Instability, we need to connect the suboptimality measured by this metric with the uncertainty of the value function estimation. We thus incorporate a novel decomposition of the planner and the agents in the algorithm design (Section 4.2).

In Section 4.3 we summarize our proposed Algorithm 1 which serves as a meta stereotype and can readily incorporate various existing RL methods.

## 4.1 Optimistic Estimation of Rewards

Note that we do not directly observe the rewards defined in (1) and have no unbiased estimates of them, so we cannot explicitly construct their optimistic estimates. However, thanks to the nature of the stable matching being a max-weight matching, we show that we can still obtain useful optimistic rewards estimates based on the optimistic estimates of the utilities.

To see this, recall the definition of reward in (1). We know that there exists some stable matching $(X_h, \tau_h)$ that maximizes $r_h$ and can be obtained by solving a linear program and its dual program (Shapley and Shubik, 1971). Define the following linear program $\mathcal{LP}(I, J, u, v)$:

$$\max_{w \in \mathbb{R}^{|I| \times |J|}} \sum_{(i,j) \in I \times J} w_{i,j}\left[u(i,j) + v(i,j)\right]$$

$$\text{s.t. } \sum_{j \in J} w_{i,j} \leq 1, \forall\, i \in I,$$

$$\sum_{i \in I} w_{i,j} \leq 1, \forall\, j \in J, \quad (4)$$

$$w_{i,j} \geq 0, \forall (i,j) \in I \times J,$$

and its dual program $\mathcal{DP}(I, J, u, v)$:

$$\min_{p:I \cup J \to \mathbb{R}^+} \sum_{a \in I \cup J} p(a) \quad \text{s.t.} \quad p(i) + p(j) \geq u(i,j) + v(i,j), \forall (i,j) \in I \times J. \tag{5}$$

Shapley and Shubik (1971) proved that the stable matching $(X, \tau)$ corresponds to the solution to the linear program (4) (for $X$) and its dual program (5) (for $\tau$). It is clear from (4) that the optimal value of the objective function is equal to the total social welfare of the stable matching. Now, suppose we have some optimistic estimates of the utilities, i.e., $\widehat{u}$ and $\widehat{v}$ such that $\widehat{u}(\cdot, \cdot) \geq u(\cdot, \cdot)$ and $\widehat{v}(\cdot, \cdot) \geq v(\cdot, \cdot)$. It is easy to see that when substituting $(\widehat{u}, \widehat{v})$ into the linear program (4), the resulting optimal value will be an upper bound of the original optimal value (see Lemma C.2 and its proof).

Based on this observation, let us return to the reward in (1). The previous argument implies that as long as we have optimistic estimates of the utilities $u_h$ and $v_h$, we can get optimistic estimates of the reward by solving the max-weight matching over the optimistic utilities. Moreover, it is further an upper bound of the following pseudo-reward:

$$\overline{r}_h(C_h, I_h, J_h, e_h) := \max_{(X_h, \tau_h) \in \mathcal{M}_h} r_h(C_h, I_h, J_h, e_h, X_h, \tau_h), \tag{6}$$

where $\mathcal{M}_h := \mathcal{M}(I_h, J_h, u_h, v_h, C_h, e_h)$ denotes the set of all myopic stable matching on $(I_h, J_h)$ with utility functions $u_h(C_h, e_h, \cdot, \cdot)$ and $u_h(C_h, e_h, \cdot, \cdot)$. Finally, the optimistic estimates of the utilities can be constructed from noisy observations of agents' utilities via any standard approach in the online learning literature.

The definition of the pseudo-reward in (6) provides a way to decompose the total regret into the planner's regret and the agents' regret, as will be clear in the next subsection.

## 4.2 Decomposition of The Planner and The Agents

Recall that we require the matching in each step to be stable, which is an additional constraint apart from maximizing the social welfare. We need to separate these two entangled goals from each other. Indeed, we will show that the total regret consists of two parts: 1) the suboptimality of the planner's policy *over the entire episode*, and 2) the distance between the proposed matching and the optimal myopic stable matching *at each step*. We identify the former as the planner's problem and the latter the agents' problem.

**The Planner's Problem.** The planner's problem focuses on the transition of the contexts, so we need to partial out the effects from the actual matching. This has been done in the definition of the pseudo-reward in (6), and the corresponding pseudo-value function $\overline{V}_h^\pi$ for $h \in [H]$ is defined as

$$\overline{V}_h^\pi(s) := \mathbb{E}_\pi \left[ \sum_{l=h}^H \overline{r}_l(s_l, e_l) \;\middle|\; s_h = s, \; e_l \sim \pi_l(\cdot|s_l), \; s_{l+1} \sim \mathbb{P}_l(\cdot|s_l, e_l), \forall h \leq l \leq H \right], \tag{7}$$

where we slightly abuse the notation $e_l \sim \pi_l(\cdot|s_l)$. Also note that we can write $s_{h+1} \sim \mathbb{P}(\cdot|s_h, e_h)$ instead of the more general $s_{h+1} \sim \mathbb{P}(\cdot|s_h, a_h)$ since we condition on $(I_h, J_h)$ and the transition of $C_h$ only depends on the planner's action $e_h$ as $C_{h+1} \sim \mathbb{P}_h(\cdot|C_h, e_h)$.

Clearly, $\overline{V}_h^\pi$ is an upper bound of $V_h^\pi$ and does not depend on the actual matching $\{X_h, \tau_h\}_{h \in [H]}$ since it has been maximized out. Now, we specify the planner's problem as trying to maximize the pseudo-value $\overline{V}_1^\pi$, and define the planner's regret given the initial state $s_1$ as

$$R^P(K) := \sum_{k=1}^K \left[ \max_\pi \overline{V}_1^\pi(s_1) - \overline{V}_1^{\pi_k}(s_1) \right] = \sum_{k=1}^K \left[ \overline{V}_1^\star(s_1) - \overline{V}_1^{\pi_k}(s_1) \right]. \tag{8}$$

From a control-theoretic perspective, the planner's problem can be viewed as learning an MDP with the same state space $\mathcal{S}$, the action space reduced to $\Upsilon$, and reward being the value of the myopic max-weight matching at each step. The reward cannot be observed, nor do we have an unbiased estimator. From an economic perspective, the planner's problem captures only the market context and not the specific market outcome (i.e. matching). Now, note that

$$R(K) = \underbrace{\sum_{k=1}^K \left[ \max_{\pi \in \Pi} V_1^\pi(s) - \overline{V}_1^{\pi_k}(s_1) \right]}_{\text{(Planner's regret)}} + \underbrace{\sum_{k=1}^K \left[ \overline{V}_1^{\pi_k}(s_1) - V_1^{\pi_k}(s_1) \right]}_{\text{(Utility difference)}},$$

where the planner's regret has been captured in (8), and it remains to control the utility difference on the agents' side.

**Agents' Problem.** The agents' problem amounts to controlling the suboptimality of each implemented matching, which boils down to SI proposed by Jagadeesan et al. (2021).

**Definition 4.1** (Subset Instability, Jagadeesan et al. 2021). *Given any agent sets $I$, $J$ and utility functions $u, v : I \times J \to \mathbb{R}$, the Subset Instability $\mathrm{SI}(X, \tau; I, J, u, v)$ of the matching and transfer $(X, \tau)$ is defined as*

$$\max_{I' \times J' \subseteq I \times J} \Big[ \Big( \max_{X'} \sum_{i \in I'} u(i, X'(i)) + \sum_{j \in J'} v(X'(j), j) \Big)$$
$$- \sum_{i \in I'} \big( u(i, X(j)) + \tau(i) \big) - \sum_{j \in J'} \big( v(X(j), j) + \tau(j) \big) \Big],$$

*where $X(\cdot)$ and $X'(\cdot)$ denotes the matched agent in matching $X$ and $X'$ respectively.*

Subset Instability has several key properties for learning. Importantly, it can be shown that given $(I, J, u, v)$, the utility difference between the optimal matching-transfer pair and $(X, \tau)$ is upper bounded by $\mathrm{SI}(X, \tau; I, J, u, v)$. With a slight abuse of notation, for $s_h = (C_h, I_h, J_h)$ and $a_h = (e_h, X_h, \tau_h)$, we denote by $\mathrm{SI}(s_h, a_h, u_h, v_h)$ the SI of $(X_h, \tau_h)$ given $I_h, J_h$ and $u_h(C_h, e_h, \cdot, \cdot)$ and $u_h(C_h, e_h, \cdot, \cdot)$. We define the regret of the agents as

$$R^M(K) \coloneqq \sum_{k=1}^{K} \mathbb{E}_{\pi_k} \Big[ \sum_{h=1}^{H} \mathrm{SI}(s_h, a_h, u_h, v_h) \Big]. \tag{9}$$

Moreover, SI itself can be bounded by the sum of optimistic bonuses. Therefore, quite surprisingly, we can control the planner's regret and the agents' regret at the same time by bounding the bonus sums. In this way, the total regret can be controlled due to the following proposition.

**Proposition 4.2** (Proof in Appendix D.1). *For $R(K)$, $R^P(K)$ and $R^M(K)$ defined by (3), (8), (9), it holds that $R(K) \le R^P(K) + R^M(K)$.*

### 4.3 A Meta Algorithm

Now, we are ready to present our meta algorithm as displayed in Algorithm 1. As is clear from the previous derivations, it suffices to construct optimistic estimates of the utilities, which then induces 1) matchings of agents and 2) optimistic estimates of the value functions. The latter then enables the optimistic planning for the planner.

In particular, for the estimation part, Algorithm 1 first constructs the Q-function estimates in a backward fashion. Using these estimates, Algorithm 1 computes optimistic estimates of utilities by calling the subroutine `UE` (utility estimation), which then leads to estimates of the pseudo-reward using the subroutine `RE` (reward estimation). Next, optimistic estimates of the Q functions are obtained via the subroutine `QE` (Q-function estimation). Next, for the planning part, Algorithm 1 chooses action in $\Upsilon$ greedily and the matching-transfer pair by calling `OM` (optimal matching). Finally, `OM` finds a matching-transfer pair which is stable with respect to the estimated utility functions.

As discussed in Section 4.1, the optimistic estimates of the rewards come from solving the linear program in (4) and its dual program in (5), which produce the optimal matching given the set of participating agents and utilities. Therefore, the `RE` oracle is defined as Algorithm 2, and the `OM` oracle is defined as Algorithm 3. The remaining subroutines (`UE` and `QE`) are flexible and can be carefully calibrated for different model assumptions. The modular nature of Algorithm 1 facilitates incorporation of existing RL algorithms. In particular, we study a special case of linear function approximation in the next section, where we provide explicit oracles for these subroutines, and show that the corresponding algorithm enjoys a sublinear regret.

## 5 Case Study: Markov Matching Markets with Linear Features

In this section, we illustrate the power of our framework under linear function approximation, which is the simplest case of function approximation, yet still a rich enough model.

---

**Algorithm 1** Sequential Optimistic Matching (SOM)

---

1: **Require:** $\lambda, \beta_u, \beta_V$
2: **Initialize:** $u_h^1 \equiv 1, v_h^1 \equiv 1$ and $\mathcal{D}_h^0 = \emptyset, \forall h \in [H]$
3: **for** episode $k = 1, 2, \ldots, K$ **do**
4:     Receive the initial state $s_1^k = (C_1^k, I_1, J_1)$
5:     Set $\overline{Q}_{H+1}^k \equiv 0$
6:     **for** stage $h = H, H-1, \ldots, 1$ **do**
7:         Estimate utilities $(u_h^k, v_h^k) \leftarrow \texttt{UE}(\mathcal{D}_h^{k-1}, \beta_u, \lambda)$
8:         Estimate pseudo-reward $\overline{r}_h^k \leftarrow \texttt{RE}(u_h^k, v_h^k, I_h, J_h)$
9:         Estimate Q-function $\overline{Q}_h^k \leftarrow \texttt{QE}(\mathcal{D}_h^{k-1}, \overline{r}_h^k, \overline{Q}_{h+1}^k, \beta_V, \lambda)$
10:    **end for**
11:    **for** stage $h = 1 \ldots, H$ **do**
12:        Planner takes action $e_h^k \leftarrow \text{argmax}_{e \in \Upsilon} \overline{Q}_h^k(s_h^k, e)$
13:        Compute the optimal matching $(X_h^k, \tau_h^k) \leftarrow \texttt{OM}(u_h^k, v_h^k, I_h, J_h, C_h^k, e_h^k)$
14:        Implement matching $(X_h^k, \tau_h^k)$
15:        Observe utilities $u_h^k(i, j), v_h^k(i, j)$ for $(i, j) \in X_h^k$
16:        Receive next state $s_{h+1}^k = (C_{h+1}^k, I_{h+1}, J_{h+1})$
17:        Update utility dataset $\mathcal{D}_h^k = \mathcal{D}_h^{k-1} \cup \{C_h^k, e_h^k\} \cup \{u_h^k(i, j), v_h^k(i, j)\}_{(i,j) \in X_h^k}$
18:    **end for**
19: **end for**

---

## 5.1 Model Assumptions

**Utility Model.** We assume there are known feature mappings $\psi : \mathcal{C} \times \Upsilon \to \mathbb{R}^d$ and $\phi : \mathcal{I} \times \mathcal{J} \to \mathbb{R}^d$, such that for any $h \in [H]$ and $(C_h, e_h, i, j)$, the utility functions are

$$u_h(C_h, e_h, i, j) = \langle \text{vec}(\psi(C_h, e_h)\phi(i, j)^\top), \boldsymbol{\theta}_h \rangle,$$
$$v_h(C_h, e_h, i, j) = \langle \text{vec}(\psi(C_h, e_h)\phi(i, j)^\top), \boldsymbol{\gamma}_h \rangle.$$

Here $\{\boldsymbol{\theta}_h, \boldsymbol{\gamma}_h\}_{h=1}^H$ are unknown parameters in $\mathbb{R}^{d^2}$. We further define the vectorized feature vector:

$$\boldsymbol{\Phi}(C_h, e_h, i, j) := \text{vec}(\psi(C_h, e_h)\phi(i, j)^\top) \in \mathbb{R}^{d^2}.$$

Then the immediate reward defined in (1) can be written as

$$r_h(s_h, a_h) = \left\langle \sum_{(i,j) \in X_h} \boldsymbol{\Phi}(C_h, e_h, i, j), \boldsymbol{\theta}_h + \boldsymbol{\gamma}_h \right\rangle.$$

**Transition Model.** Conditioning on the agents' sets $\{I_h, J_h\}_{h=1}^H$, the state transition reduces to that of the contexts. We assume a linear transition model (Jin et al., 2020):

$$\mathbb{P}_h(C_{h+1}|C_h, e_h) := \langle \psi(C_h, e_h), \boldsymbol{\mu}_h(C_{h+1}) \rangle, \tag{10}$$

for all $h \in [H]$, where $\boldsymbol{\mu}_h : \mathcal{C} \to \mathbb{R}^d$ is some unknown measure.

Next, we introduce some standard assumptions for matching and linear MDPs (Jin et al., 2020; Jagadeesan et al., 2021).

**Assumption 5.1.** We assume WLOG that for any $h \in [H]$, $\|u_h(\cdot)\|, \|v_h(\cdot)\| \leq 1$. Assume that for any $(I_h, J_h) \in 2^{\mathcal{I}} \times 2^{\mathcal{J}}$, there exists $W_h > 0$ such that for any context $C \in \mathcal{C}$ and action $e \in \Upsilon$, the max-weight (possibly unstable) matching on $(I_h, J_h)$ with utility functions $u_h(C, e, \cdot, \cdot)$ and $v_h(C, e, \cdot, \cdot)$ has total utility upper bounded by $W_h$.

The quantities $\{W_h\}_{h=1}^H$ can be viewed as a measure of complexity of the matching problem, and indeed $\sum_{h=1}^H W_h$ determines the magnitude of $V_1^{\pi^\star}$. Assumption 5.1 implies a trivial upper bound that $W_h \leq \min\{|I_h|, |J_h|\}$, but when the max-weight matching involves only a subset of the agents, $W_h$ can be much smaller. Therefore, we regard $\{W_h\}_{h=1}^H$ as instance-dependent parameters.

| **Algorithm 2** Reward Estimation (RE) | **Algorithm 3** Optimal Matching (OM) |
|---|---|
| **Input:** $u, v, I, J$ | **Input:** $u, v, I, J, C, e$ |
| **Output:** $\widehat{r}(C, I, J, e)$ as the solution to the $\mathcal{LP}(I, J, u, v)$, for any $(C, e) \in \mathcal{C} \times \Upsilon$ | **Output:** $(X, \tau)$ as the solution of the primal-dual program defined by (4): $\mathcal{LP}(I, J, u(C, e, \cdot, \cdot), v(C, e, I, J))$, and (5): $\mathcal{DP}(I, J, u(C, e, \cdot, \cdot), v(C, e, I, J))$. |

**Assumption 5.2.** We assume WLOG that $\|\boldsymbol{\psi}(C, e)\|_2 \leq 1$ and $\|\boldsymbol{\phi}(i, j)\|_2 \leq 1$, implying $\|\boldsymbol{\Phi}(C, e, i, j)\|_2 \leq 1$, for any $(C, e, i, j)$. We assume that for any $h \in [H]$, $\|\boldsymbol{\theta}_h\|_2 \leq d$, $\|\boldsymbol{\gamma}_h\|_2 \leq d$ and $\|\boldsymbol{\mu}_h(\cdot)\|_2 \leq \sqrt{d}$. Moreover, assume that $\max_{h \in [H]} \|\int_{\mathcal{C}} f(C) \mathrm{d}\mu_h(C)\|_2 \leq \sqrt{d}$ for any function $f : \mathcal{C} \to \mathbb{R}$ such that $\sup |f| \leq 1$.

**Assumption 5.3.** We assume that the observed utilities of matched pairs are the true utilities plus independent 1-subgaussian noise.

## 5.2 Algorithms

Based on previous model assumptions, we now present explicit computation oracles for Algorithm 1.

**Utility Estimation.** At the beginning of episode $k$, for any $h \in [H]$, denote the available data by $\mathcal{D}_h^{k-1}$ which consists of $\{u_h^t(i, j)\}_{t \in [k-1]}^{(i,j) \in X_h^t}$ and $\{v_h^t(i, j)\}_{t \in [k-1]}^{(i,j) \in X_h^t}$, where by default $\mathcal{D}_h^0 = \emptyset$. For the linear case, each $u_h^t(i, j) = \langle \boldsymbol{\Phi}(C_h^t, e_h^t, i, j), \boldsymbol{\theta}_h \rangle + noise$, and similar for $v_h^t(i, j)$. So we can estimate $\boldsymbol{\theta}_h$ and $\boldsymbol{\gamma}_h$ by ridge regression:

$$\boldsymbol{\theta}_h^k = (\boldsymbol{\Sigma}_h^k)^{-1} \sum_{t=1}^{k-1} \sum_{(i,j) \in X_h^t} \boldsymbol{\Phi}(C_h^t, e_h^t, i, j) u_h^t(i, j),$$

$$\boldsymbol{\gamma}_h^k = (\boldsymbol{\Sigma}_h^k)^{-1} \sum_{t=1}^{k-1} \sum_{(i,j) \in X_h^t} \boldsymbol{\Phi}(C_h^t, e_h^t, i, j) v_h^t(i, j), \tag{11}$$

$$\boldsymbol{\Sigma}_h^k = \lambda \mathbf{I}_{d^2} + \sum_{t=1}^{k-1} \sum_{(i,j) \in X_h^t} \boldsymbol{\Phi}(C_h^t, e_h^t, i, j) \boldsymbol{\Phi}(C_h^t, e_h^t, i, j)^\top.$$

We then add a bonus to ensure optimism in the utility function estimates and apply the truncation:

$$u_h^k(C, e, i, j) = \left( \langle \boldsymbol{\Phi}(C, e, i, j), \boldsymbol{\theta}_h^k \rangle + \beta_u \|\boldsymbol{\Phi}(C, e, i, j)\|_{(\boldsymbol{\Sigma}_h^k)^{-1}} \right)_{[-1,1]},$$
$$v_h^k(C, e, i, j) = \left( \langle \boldsymbol{\Phi}(C, e, i, j), \boldsymbol{\gamma}_h^k \rangle + \beta_u \|\boldsymbol{\Phi}(C, e, i, j)\|_{(\boldsymbol{\Sigma}_h^k)^{-1}} \right)_{[-1,1]}. \tag{12}$$

The UE for the linear case is summarized in Algorithm 4.

**Q-function Estimation.** By the assumption on the context transition, for any function $f$, the function $\mathbb{P}_h f$ is linear in features $\boldsymbol{\psi}$, which induces the commonly used LSVI-type algorithms (Jin et al., 2020). Together with the reward estimates, we can estimate the Q-function using Bellman equation via backward ridge regression. For each $(k, h)$, denote the estimate of the Q-function by $\overline{Q}_h^k$ and the value function by $\overline{V}_h^k$. Given $\overline{Q}_{h+1}^k$, maximizing over $e \in \Upsilon$ yields $\overline{V}_{h+1}^k$, then we solve the following ridge regression:

$$\mathbf{w}_h^k = \underset{\mathbf{w} \in \mathbb{R}^d}{\arg\min} \sum_{t=1}^{k-1} \left[ \overline{V}_{h+1}^k(C_{h+1}^t) - \boldsymbol{\psi}(C_h^t, e_h^t)^\top \mathbf{w} \right]^2 + \lambda \|\mathbf{w}\|_2^2,$$

which further yields the estimated expectation of $\overline{V}_{h+1}^k$:

$$\widehat{\mathbb{P}}_h \overline{V}_{h+1}^k(\cdot, \cdot) = \boldsymbol{\psi}(\cdot, \cdot)^\top \mathbf{w}_h^k + \beta_V \|\boldsymbol{\psi}(\cdot, \cdot)\|_{(\boldsymbol{\Lambda}_h^k)^{-1}}, \tag{13}$$

where $\boldsymbol{\Lambda}_h^k = \sum_{t=1}^{k-1} \boldsymbol{\psi}(C_h^t, e_h^t) \boldsymbol{\psi}(C_h^t, e_h^t)^\top + \lambda \mathbf{I}_d$. Finally, we estimate $Q_h^k$ using the Bellman equation. These steps are summarized in Algorithm 5.

| **Algorithm 4** Utility Estimation (UE) | **Algorithm 5** Q-function Estimation (QE) |
|---|---|
| **Input:** $\mathcal{D}_h^{k-1}, \beta_u, \lambda$ | **Input:** $\mathcal{D}_h^{k-1}, \overline{r}_h^k, \overline{Q}_{h+1}^k, \beta_V, \lambda$ |
| **if** $\mathcal{D}_h^{k-1}$ is empty **then** | $\overline{V}_{h+1}^k(C, I_h, J_h) = \max_e \overline{Q}_{h+1}^k(C, e, I_h, J_h)$ |
|    **Output**: $u_h^k \equiv 1$ and $v_h^k \equiv 1, \forall h \in [H]$ | Compute $\widehat{\mathbb{P}}_h \overline{V}_{h+1}^k$ by (13) |
| **end if** | $\overline{Q}_h^k(C, I_h, J_h, e) = (\overline{r}_h^k(C, I_h, J_h, e) +$ |
| Compute $\boldsymbol{\theta}_h^k, \boldsymbol{\gamma}_h^k$ by (11) | $\widehat{\mathbb{P}}_h \overline{V}_{h+1}^k(C, e))_{[0, \sum_{l=h}^H W_l]}$ |
| Estimate utility functions with $u_h^k$ and $v_h^k$ by (12) | |
| **Output:** the functions $u_h^k$ and $v_h^k$ | **Output:** function $\overline{Q}_h^k$ |

## 5.3 Theoretical Results

In this section, we present our theoretical results, with the proofs deferred to Appendix D. We start with the main theorems on the agents' regret and the planner's regret.

**Theorem 5.4** (Agents' Regret). *With probability at least $1 - 2\delta$, the agents' regret can be bounded as $R^M(K) \leq \mathcal{O}(d^2(\sum_{h=1}^H \min\{|I_h|, |J_h|\})\kappa\sqrt{K})$ where $\kappa = \log(dK \min(|\mathcal{I}|, |\mathcal{J}|)/\delta)$.*

Comparing this result with Theorem 5.3 in Jagadeesan et al. (2021), which proves the regret of their `MatchLinUCB` algorithm designed for the linear utility class under $H = 1$, and ignoring the logarithmic term, we see that both have a linear dependence on the cardinality of the agents' set, while we have an extra summation over the horizon $H$ due to the sequential setting. It might seem that their $d$-dependence is $\mathcal{O}(d)$ while ours is $\mathcal{O}(d^2)$, but this is because their feature is in $\mathbb{R}^d$ while our feature $\Phi$ is $d^2$-dimensional. Therefore, the dominant term in our regret bound matches that of Jagadeesan et al. (2021), and our result can be viewed as an extension.

**Theorem 5.5** (Planner's Regret). *Under Assumption 5.1, 5.2, assuming $KH > 32$, there exists a problem-independent constant $\eta > 0$, such that for any $\delta > 0$, setting $\lambda = 1$, $\beta_V = \eta d^2(\sum_{h=1}^H W_h) \cdot \sqrt{\iota}$ where $\iota = \log(dKH \min\{|\mathcal{I}|, |\mathcal{J}|\}/\delta)$ and $\beta_u$ as given in Lemma C.1, the planner's regret is bounded by $R^P(K) \leq \mathcal{O}(\eta d^{5/2} H(\sum_{h=1}^H W_h)\iota\sqrt{K})$ with probability at least $1 - \delta$.*

Combining the above two theorems, we have shown that we find the optimal policy for both the planner and agents at sublinear rates, and accordingly for the total regret. Notably, the regret upper bound only depends on the size of the market through $\{W_h\}_{h=1}^H$, which is instance-dependent. Due to the sublinear regret, our algorithm can be further adapted to a PAC algorithm (cf. Jin et al., 2018). We also remark that our result with LSVI-type estimation can be naturally extended to Eluder dimension (Wang et al., 2020; Ayoub et al., 2020).

The key step in our analysis is to show that the estimated pseudo-reward function $\overline{r}_h^k$ satisfies optimism, i.e., $\overline{r}_h^k \geq \overline{r}_h$, and we need to ensure that $\overline{r}_h^k$ is not too far away from $\overline{r}_h$ (Lemma C.2). Due to space limit, we refer interested readers to Appendix C for a proof sketch of the main theory and an introduction of all the key technical lemmas. The full proof is presented in Appendix D.

# 6 Conclusion

We propose a novel Markov matching market model and a general framework that incorporates max-weight matching and RL algorithms for efficient online learning. We prove that our proposed algorithms achieve sublinear regret under proper structural assumptions on the structural model. Overall, our algorithm addresses the coupled challenges of sequential exploration, matching stability, and function approximation.

## Acknowledgments and Disclosure of Funding

Zhaoran Wang acknowledges National Science Foundation (Awards 2048075, 2008827, 2015568, 1934931), Simons Institute (Theory of Reinforcement Learning), Amazon, J.P. Morgan, and Two Sigma for their supports.

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
