# OpenReview forum: "Learn to Match with No Regret: Reinforcement Learning in Markov Matching Markets"
_NeurIPS.cc/2022/Conference — NeurIPS 2022 Accept_

### Official Review · Reviewer_UFsj · 2022-07-09

**Rating:** 6
**Confidence:** 3
**Soundness:** 3 good
**Presentation:** 3 good
**Contribution:** 2 fair

**Summary:**

This work frames a Markov matching market model and tackles it in a reinforcement learning (RL) framework. The model extends the previous works (e.g., [29]) to consider a dynamic scenario. Techniques from both RL (i.e., optimism) and market matching (i.e., max-weight matching) are connected to solve this problem.

A general framework is first proposed, which nicely decomposes the problem into two sides (the planner and the agent) by introducing the pseudo-reward. Both sides depend on optimistic estimations of the user's utilities. The planner's problem becomes an RL one, where the rewards are not directly observed but are instead estimated by solving the stable matching with optimistic estimations (through an LP). The agent's problem boils down to finding a good matching with bandit feedback, which again is based on optimistic estimations. In the case of a linear model, sublinear regret bounds are proved.

**Questions:**

1. I have only skimmed through the proof, and it would be nice if the authors can highlight some key theoretical challenges in this work, especially compared to [33] (for the planner's side) and to [29] (for the agents' side).
2. It would be helpful if some clarifications (or practical motivations) can be given on why the transition does not depend on the selected matching and transfer, which seems critical for regret decomposition.
3. In Eqn. (6), I am wondering about the definition of $\mathcal{M}_h$. Does the illustration on line 203 mean that there are many myopic stable matchings and they lead to different rewards? It confuses me given Proposition B.2, which says the stable matching maximizes the reward.

**Limitations:**

See weakness.

**Strengths And Weaknesses:**

- (S) This work connects the study of RL and economic problems (i.e., market matching), which is interesting. I believe this is a good attempt to benefit other fields from the recent advances in RL theory.
- (S) The design and analysis are sound and enlightening. Especially, techniques from RL and market matching are nicely combined.
- (S) The overall writing is clear under the page limits. Sufficient background knowledge of market matching is provided in the appendix (which benefits me a lot).
- (S) One especially interesting point in my view is the usage of the optimistic estimation of user utilities. It plays an important role on both sides of the problem. For the planner's side, these estimations ensure an optimistic version of pseudo-reward, which naturally provides the required optimism in RL learning (thus confidence bound is only added to the estimation of the transition in the linear model). For the agents' side, it naturally links the problem back to [29] to find good matching with bandit feedback.
- (W) As I am unfamiliar with the literature on market matching, the techniques are somewhat not exciting from the RL theory perspective. Especially, after decomposing the problem into two sides, the planner's side (Theorem 5.5) is similar to the standard RL setting (thus solved by similar techniques in [33]). The agents' side seems to closely follow [29] (while I am not familiar with it), especially from Theorem 5.4. Thus, the overall feeling about this work is adding a layer (i.e., the planner) to the market matching problem to handle the dynamic environment.
- (W) Following the previous point, I believe in general, a more clear statement of theoretical contributions (especially, insights) should be given to make the paper stronger given its theoretical nature.
- (W) The decomposition of the regret seems to heavily really on the assumption that only the actions of the planner affect the transition, while the selected matchings and transfers only affect reward in each step. I am not sure whether this is a reasonable assumption in economic literature. However, up to my understanding, it largely simplifies the scenario.

---

> ### Author Response · Authors · 2022-08-02
> **Response to Reviewer UFsj**
>
> Thank you so much for your valuable feedback! Please see our response below.
>
> **Q1**: As I am unfamiliar with the literature on market matching, the techniques are somewhat not exciting from the RL theory perspective ... Thus, the overall feeling about this work is adding a layer (i.e., the planner) to the market matching problem to handle the dynamic environment.
>
> **A1**:
> We would like to emphasize that our work has significant differences compared to both online RL and matching bandit in terms of problem setting, algorithm design and technical analysis.
>
> - Compared to online RL (e.g. [33]), in our problem setting the reward signal is missing, and even an unbiased estimator of the true reward is not available. The reason is that the true reward is related to the solution to a linear programming since we are studying optimal matching.
> - Due to this reason, we need to use Subset Instability as the suboptimality measure, which brings unique technical challenges. Specifically, Subset Instability is a nonlinear functional of the rewards of the agents. How to do uncertainty quantification of the value function estimation and regret decomposition in the presence of such a suboptimality measure was never studied before in the online RL literature. Furthermore, our regret decomposition does not depend on the actual form of the state transition and thus is applicable in general and of independent technical interest.
> - Compared to matching bandits (e.g. [29]), bandit algorithms cannot handle Markovian structure (i.e. state transition). For the analysis, the bandit setting does not need to consider the propagation of error in terms of $H$ as we do.
>
> Therefore, the technique of our work is very different from that of [29] and [33].
>
> ---
>
> **Q2**: Highlight some key theoretical challenges in this work: compared to [33] (for the planner's side) and to [29] (for the agents' side).
>
> **A2**: We face unique technical challenges compared to standard RL and matching bandits.
>
> Compared with standard MDPs in RL([33]):
> - We aim to maximize the social welfare function and thus the regret involves a new notion called subset instability, which is a nonlinear functional of the rewards of the agents.
> - Due to the above reason, the standard RL algorithm cannot directly apply here and we need to solve a sequence of combinatorial optimization problems to obtain a UCB estimate.
> - Moreover, as the regret is in terms of the subset instability, we need to connect the suboptimality measured by this metric with the uncertainty of the value function estimation, which requires new analyses.
>
> Compared to matching bandits ([29]):
> - Due to the Markovian structure with state transition, we face an extra problem of estimating the reward of optimal matching at each step $h$. This does not exist in [29].
> - We need to handle the propagation of error in terms of $H$.
>
> **Reference**:
>
> [29] Learning Equilibria in Matching Markets from Bandit Feedback, Jagadeesan, M., Wei, A., Wang, Y., Jordan, M. I. and Steinhardt, J., Neurips 2021.
>
> [33] Provably Efficient Reinforcement Learning with Linear Function Approximation, Chi Jin, Zhuoran Yang, Zhaoran Wang, Michael I. Jordan, COLT 2020
>
> ---
>
> **Q3**: The decomposition of the regret seems to heavily rely on the assumption that only the actions of the planner affect the transition, while the selected matchings and transfers only affect reward in each step.
>
> **A3**: Technically speaking, our current framework could allow the matchings and transfers to affect the transition by extending the feature mapping to also involve the matchings and transfers. To do so, we can modify the feature mapping $\mu_h$ to be $\mu_(C_h, X_h, \tau_h)$, and the corresponding state transition becomes $\mathbb{P}_h (C_{h+1} | C_h, X_h, \tau_h)$.
>
> Furthermore, very importantly, our regret decomposition technique does not depend on the actual form of the state transition. Therefore our analysis still holds if we allow the matchings and transfers to affect the transition.
>
> We did not choose to involve the matchings and transfers in the transition function because we want to simplify the notation and ease the presentation of the result.
>
> ---
>
> **Q4**: In Eq. (6), I am wondering about the definition of $\mathcal{M}_h$. Does the illustration on line 203 mean that there are many myopic stable matchings and they lead to different rewards? It confuses me given Proposition B.2, which says the stable matching maximizes the reward.
>
> **A4**: You are correct: any stable matching is max-weight matching according to [Shapley and Shubik, 1955].
> We are very sorry for the confusion caused by Eq.(6): although Eq. (6) is not wrong, it is a bit misleading. We should simply say: let $\mathcal{M}_h$ denote the set of all matching-transfer pairs on $(\mathcal{I}_h, \mathcal{J}_h)$. Thank you very much for pointing this out and we will modify it in the revision.

---

> > ### Comment · Reviewer_UFsj · 2022-08-08
> > **Thanks for the responses**
> >
> > I would like to thank the authors for the feedbacks. I still have positive opinions about this work, however my major concerns are not fully addressed.
> > - Given the authors' feedbacks, I still believe this work is just combining RL and market matching. For the planner's side, an optimistic estimation of the reward is enough although we do not have an unbiased estimator. For the agent's side, the idea is essentially that of the matching bandits. Maybe the major technical novelty of this work is that it nicely connects the planner's side and the agent's side via the optimistic reward estimation. This allows the adding a layer of RL transitions to the top of matching bandits.
> > - I do not think it is trivial to let the selected matchings and transfers influence the state transitions. The current approach is to have the matching and transfers selected myopically in each step with rewards estimations for that step. However, if the matching selection affects future states, my intuition is that it should not be selected myopically with the current rewards. Hopefully the authors can elaborate more on this point. Given this concern, I think the current combination of RL and market matching is less interesting.

---

> > > ### Author Response · Authors · 2022-08-08
> > > **Thank you for the feedback!**
> > >
> > > Thank you for your feedback!
> > >
> > > ---
> > > The reviewer is correct that we do not need an unbiased estimator for the planner’s reward since an optimistic estimation suffices. We also thank the reviewer for recognizing that connecting the planner side and the user side via optimism is technically novel.
> > >
> > > We think connecting the planner side and the user side via optimism is part of our technical novelty. Another part is to bound the cumulative Subset Instability. Note that this cumulative SI involves taking expectation with respect to the randomness of the trajectory. This is still different from the static bandit setting [Jagadeesan et al, 2021], where the SI does not accumulate across different steps.
> > >
> > > ---
> > >
> > > We will clearly explain ‘let the selected matchings and transfers influence the state transitions’ as follows.
> > > Technically, we mean that we only redefine the feature mapping to be $\mu_h(C_h,X_h,\tau_h)$, and the corresponding state transition becomes $\mathbb{P}(C_{h+1}$ $| C_h$, $X_h, \tau_h)$. But we do NOT change the definition of regret, i.e., we still compare with the optimal myopic matching, as we currently defined by Eq. 8 in our paper. In other words, our regret definition does not look into the future.
> > >
> > > The reviewer is indeed correct that, if we allow the regret definition to look into the future, then the reviewer is correct that an optimal matching is likely not the optimal action. We totally agree that this would be a very interesting setting.
> > >
> > > Still, we believe our myopic setting is also very meaningful. (1) myopic agents is a well-motivated setting; (2) it is also technically challenging because the optimization problem is a bilevel optimization (i.e. find the max-weight matching; maximize the planner’s regret).
> > >
> > > Extension to non-myopic agents is a very ambitious problem and would be a future direction.

---

> > > > ### Comment · Reviewer_UFsj · 2022-08-09
> > > > **Thanks for the feedback.**
> > > >
> > > > I would like to thank the authors again for the feedback. I have no other concerns.

---

### Official Review · Reviewer_kFzA · 2022-07-09

**Rating:** 7
**Confidence:** 4
**Soundness:** 3 good
**Presentation:** 3 good
**Contribution:** 3 good

**Summary:**

This paper considers Markov Matching Markets consisting of three parties: two sides of agents (e.g. sellers and buyers) and a principal. They extend the setting of matching markets with bandit feedback to a problem where the principal, next to choosing the matching, determines context transitions (which are Markov). The authors address this problem treating it inside the RL/MDP framework. The algorithm Sequential Optimistic Matching is proposed and a sub-linear regret bound proven for the special case of linear function approximation.

**Questions:**

To me it was not entirely clear how you obtain optimistic estimates of the utilities and also why the added bonuses in equation (12) are the right choice. Could you give an explanation/intuition of the chosen bonuses and how you obtain the optimistic estimates?


Minor suggestions:

1. Please define the norm $\lVert \cdot \rVert_{A}$, where $A$ is a matrix, somewhere in the text. The definition of this norm was not clear (to me) when it was first used in equation (12). (I also didn't find a definition in the appendix, but had to check in Abbasi-Yadkori et al.)

2. In Section 4.3, it would be helpful to state that the subroutines UE, RE, QE, OM are described in Section 5. Also writing the names out directly after first using the abbreviation would help (in particular for QE and OM).


**Limitations:**

As mentioned above, I think that the paper would benefit from a more extensive discussion of the results and limitations. In particular, only the case of linear function approximation was analysed in the paper and additional comments on this would be helpful.

**Strengths And Weaknesses:**

The problem considered in this paper is interesting and the chosen problem formulation intuitive and well-motivated. To the best of my knowledge, the work is novel and has not been studied in this form before. Moreover, the formulation of this dynamics matching problem as a RL problem is quite nice and informative.
I also found the presentation in the first four sections exceptionally well done considering the amount of notation and concepts the paper is forced to introduce (except for some minor points, see below).

I think that the paper would benefit from a more extensive discussion of the results. In particular, as only linear function approximation is discussed, additional comments on this would be helpful to include in the paper. I understand that the page limit made this difficult to add in the main paper.

(Although I had a look into the proofs, I did not rigorously check for their correctness.)

---

> ### Author Response · Authors · 2022-08-02
> **Response to Reviewer kFzA**
>
> Thank you so much for your valuable feedback! Please see our response below.
>
> **Q1**: To me it was not entirely clear how you obtain optimistic estimates of the utilities and also why the added bonuses in equation (12) are the right choice. Could you give an explanation/intuition of the chosen bonuses and how you obtain the optimistic estimates?
>
> **A1**:
> Note that by our assumption in Section 5.1, the utilities are linear functions of the known feature mappings, so it follows from the standard approach in linear bandits to construct optimistic estimates of the utilities. Then since the optimal matching comes from solving a maximization problem over the utilities, the optimism on the utilities naturally transfers to that on the rewards, as is proven in Lemma C.2 in the appendix. This further guarantees optimistic estimates of the value functions.
>
> ---
>
> **Q2**: Minor suggestion (clarification of notations and abbreviation).
>
> **A2**: Thank you for your careful reading! We will modify all the related places in the revision. Specifically, for a vector x, the norm $||x||_A$ is defined as $||x||_A = \sqrt{x^\top A x}$. We will explain the abbreviations when first using them.
>
> ---
>
> **Q3**: I think that the paper would benefit from a more extensive discussion of the results. In particular, as only linear function approximation is discussed, additional comments on this would be helpful to include in the paper. I understand that the page limit made this difficult to add in the main paper.
>
> **A3**: Thank you for the suggestion! We study the linear case for simplicity of presentation. Our result can indeed be extended to more general cases. For example, we can consider the case where the utility function is from a neural network (NN) class. The corresponding analysis and proof need to be modified in the following way:
>
> 1. We first need to use a class of Neural Tangent Kernels (NTK) to approximate the NN class. This is because our UCB-type algorithm requires the construction of bonus terms. Using NTK to approximate the NN-type utility functions allows us to construct these bonus terms. In terms of the regret bound, doing so will add an extra approximation error from approximating NN using NTK, which decays to zero as the number of neuron goes to infinity. See, e.g., [1].
> 2. We need to replace the logarithmic covering number of the linear function class, which is $d$,  by the logarithmic covering number of the NN class.
> 3. The dimension $d$ will be replaced by the effective dimension of the NTK class. This dimension can be computed from the eigenvalues of the kernel.
>
> Both the logarithmic covering number and the effective dimension can be computed using the eigenvalue decay conditions of the NTK, which depends on the activation function and the distribution of the initial network weights.
> Therefore, our result can indeed be extended to more complicated and richer function classes. Thank you for the suggestion and we will add the above discussion in the revision.
>
> **Reference**:
>
> [1] Fine-grained analysis of optimization and generalization for overparameterized two-layer neural networks. Sanjeev Arora, Simon S. Du, Wei Hu, Zhiyuan Li, Ruosong Wang. ICML 2019.

---

> > ### Comment · Reviewer_kFzA · 2022-08-08
> > **Thank you for your response**
> >
> > Thank you very much for your detailed response.

---

### Official Review · Reviewer_GDGf · 2022-07-11

**Rating:** 8
**Confidence:** 1
**Soundness:** 4 excellent
**Presentation:** 4 excellent
**Contribution:** 4 excellent

**Summary:**

This paper proposes a Markov matching market model, and it proposes a general framework that incorporates max-weight matching and RL algorithms for efficient online learning. This paper proves that the algorithms achieve sublunar regret under proper structural assumptions.

**Questions:**


This paper is out of my area of expertise; however, I find the Markov matching market problem it solves very interesting and important. I hope the paper can provide more intuitions on the technical results and assumptions to make it more accessible to the general technical audience.

(1) Given that there is no empirical result, I wonder if the paper can provide some insights on what scenarios the method can outperform matching bandits and whether these scenarios are realistic in practice.
(2) Please provide explanations/intuitions on Assumptions 5.1 and 5.2, why they are reasonable to make, and what they mean in practice.
(3) What’s the complexity of the proposed meta-algorithm? Is this algorithm realistic to implement in practice?

**Strengths And Weaknesses:**

Strengths:
This paper studies an important matching market problem. It extends traditional online RL with a matching problem at each step and incorporates a transition between contexts into matching markets. The problem combines online RL and matching bandit. This paper provides a general analysis framework and shows that the proposed algorithm achieves sublinear regret under proper structural assumptions.

Weaknesses:
Despite the theoretical proof and contribution, the paper does not have an experiment to evaluate how the method works in practical scenarios and potentially compare with online RL and matching bandits, both of which are supposed to perform worse than this paper. Given that the paper is practically-motivated, and its contribution from the literature results from the practical “matching” problem and “dynamic transitions”, an empirical evaluation seems necessary to better ground the contribution to the literature.

---

> ### Author Response · Authors · 2022-08-02
> **Response to Reviewer GDGf (part 1)**
>
> Thank you so much for your valuable feedback! Please see our response below.
>
> **Q1**:
> In what scenarios the method can outperform matching bandits and whether these scenarios are realistic in practice.
>
> **A1**:
> Our method can outperform matching bandits for scenarios where we need to solve a sequence of dynamic matching problems with the environment (including contexts and states of the agents) evolving within certain time horizon.
>
> Indeed, the setting of matching bandits [1] does not handle such dynamic setting involving state transitions, which are quite common in real-world scenarios. A representative example of such scenarios is the ride-hailing platform like Uber and Lyft, which matches drivers with passengers ([2], [3], etc.). The state here can be, for example, the overall supply (the drivers and the price they ask) and demand (the passengers and their willingness to pay) in the ride-hailing market. This is captured by the utility functions in our model, which involve the state as a parameter.
>
> Moreover, transition of states is common in real-world scenarios, which can also be captured by our model. Specifically, our model allows the state transition to be determined by the action of the central planner (e.g. ride-hailing companies like Uber). A typical example of the action is that the company can incentivise the drivers by giving them more stipends, or incentivizes the passengers by distributing coupons. These actions can surely affect the state afterwards.
>
> Overall, our result can be viewed as a more generalized theory of certain matching bandits problems. And it is motivated by dynamic real-world environments.
>
>
> **Reference**:
> [1] Jagadeesan, M., Wei, A., Wang, Y., Jordan, M. I. and Steinhardt, J.. Learning equilibria in matching markets from bandit feedback. Neurips 2021.
>
> [2] Dynamic Type Matching. Ming Hu , Yun Zhou. Manufacturing & Service Operations Management.
>
> [3] Ride-hailing order dispatching at didi via reinforcement learning. Zhiwei (Tony) Qin , Xiaocheng Tang, Yan Jiao, Fan Zhang, Zhe Xu, Hongtu Zhu, Jieping Ye. INFORMS Journal on Applied Analytics 50 272–286.
>
>
> ---
>
> **Q2**:
> Why are Assumptions 5.1 and 5.2 reasonable, and what they mean in practice?
>
> **A2**:
> They are all standard regularity assumptions and are commonly made by representative works in linear bandits/reinforcement learning. We explain each of them in the following.
>
> The assumptions that $\|u\|$, $\|v\|$, $\|\psi\|$, $\|\phi\|\leq 1$, $\|\theta\|$, $\|\gamma\|\leq d$, $\|\mu\|\leq \sqrt{d}$ are standard normalization assumptions made by seminal works including [1], [2], [3], to mention a few. These assumptions of the upper bounds here (i.e. $\leq 1, d,$or $\sqrt{d}$) are without loss of generality: we just require the true parameters and feature mappings to have bounded norms and then apply a normalization. Without this bounded norm assumption, no theoretical guarantee can be derived for linear bandits/MDPs.
>
> The assumption that the total utility is upper bounded by $W_h$ is also a normalization assumption. It is simply the upper bound of the optimal utility. Since we are studying the difference between the utility achieved by our algorithm and the optimal utility, the optimal utility must be finite for the comparison to be valid. This is standard in matching bandits literature (e.g. [4]). In reality, a finite optimal utility is also reasonable, since the total quantity of products/services/money is usually finite. So this is a very realistic assumption. Note that our algorithm does NOT need to know this upper bound.
>
> **Reference**:
>
> [1] Provably Efficient Reinforcement Learning with Linear Function Approximation, Chi Jin, Zhuoran Yang, Zhaoran Wang, Michael I. Jordan, COLT 2020
>
> [2] Is Pessimism Provably Efficient for Offline RL? Ying Jin, Zhuoran Yang, Zhaoran Wang, ICML 2021
>
> [3] Nearly Minimax Optimal Reinforcement Learning for Linear Mixture Markov Decision Processes, Dongruo Zhou, Quanquan Gu, Csaba Szepesvari, COLT 2021
>
> [4] Learning Equilibria in Matching Markets from Bandit Feedback, Jagadeesan, M., Wei, A., Wang, Y., Jordan, M. I. and Steinhardt, J., Neurips 2021.

---

> > ### Author Response · Authors · 2022-08-02
> > **Response to Reviewer GDGf (part 2)**
> >
> > **Q3**:
> > What’s the complexity of the proposed meta-algorithm? Is this algorithm realistic to implement in practice?
> >
> > **A3**: The meta-algorithm (Alg. 1) is indeed realistic. The main computation overhead of Alg. 1 are Line 7, 8, and Line 13. Specifically:
> > - Line 8 is a linear programming (LP) for each $(C,e)$ by the description of Algorithm 2. There are many available solvers for LP (e.g., interior point method). Also, the computational complexity of solving LP is polynomial in the number of variables and constraints. Importantly, we do NOT need to compute for each $(C,e)$. We only need to compute for each action $e$ and the observed context $C_h^k$. Note that this is very important since context space is usually very large or even continuous in many cases. The reason why we only need to compute for each observed context $C_h^k$ is that in line 12 of the algorithm the planner only needs to pick the optimal action according to the Q-function estimate evaluated at the **realized contex**t. Therefore, the complexity of line 8 is then poly$(|\mathcal{I}|, |\mathcal{J}|, |\Upsilon|)$.
> > - Line 7 is a regression problem. In the linear case discussed in Section 5, Line 7 corresponds to a ridge linear regression problem and has a closed form solution. The computational complexity is poly$(d, |\mathcal{I}|, |\mathcal{J}|)$.
> > - Line 13 is a linear programming and its dual program, which also have complexity poly$(|\mathcal{I}|, |\mathcal{J}|)$.
> >
> > Therefore, all the parts of Alg. 1 are implementable in practice with linear programming solvers and regression solvers. Since the main algorithm is run for $K$ episodes and each episode contains $H$ steps, the overall computational complexity is poly$(K, H, d, |\mathcal{I}|, |\mathcal{J}|, |\Upsilon|)$ and is **independent of** $|\mathcal{C}|$.
> >
> > Thank you for pointing this out and we will add the above in the updated version of the paper.

---

> > > ### Comment · Reviewer_GDGf · 2022-08-09
> > > **Thank you very much for your detailed response. I have no further comments.**
> > >
> > > Thank you very much for your detailed response. I have no further comments.

---

### Official Review · Reviewer_zRmf · 2022-07-11

**Rating:** 6
**Confidence:** 2
**Soundness:** 4 excellent
**Presentation:** 4 excellent
**Contribution:** 4 excellent

**Summary:**

This paper extends the results of [1] to a dynamic setting. To do this, the authors first propose a new formulation - markov matching markets. This formulation can capture centralized matching problems that may arise in ride-sharing apps, for example. They propose a general framework that used optimistic value iteration + max-weight matching such that any RL algorithms can be incorporated for efficient online learning. They prove that this approach achieves sublinear regret under certain assumptions.

[1] Jagadeesan, M., Wei, A., Wang, Y., Jordan, M. I. and Steinhardt, J. (2021).
Learning equilibria in matching markets from bandit feedback

**Questions:**

N/A

**Limitations:**

There isn't a discussion of limitations in the paper.

**Strengths And Weaknesses:**

**Strengths**:
1. This paper initiates the study on dynamic matching markets through tools from machine-learning. The application of this formulation is well motivated through the example of ridesharing platforms
2. The markov matching market formulation is novel and their approach combines max-weight matching + RL. Their show that their proposed approach achieves sub-linear regret under certain assumptions. The results and formulation are a natural extension to [1] and their main novelty comes from addressing new challenges that are specific to dynamic matching markets such as propagation of error through the time steps and from complexities that arise from regret decomposition.
3. The paper is well-written and well organized. It's very clear and easy to read.

**Weakness**:
1. The evaluation seems limited - they only consider linear features. Perhaps this paper could .benefit a little from a limitations section.

[1] Jagadeesan, M., Wei, A., Wang, Y., Jordan, M. I. and Steinhardt, J. (2021).
Learning equilibria in matching markets from bandit feedback

---

> ### Author Response · Authors · 2022-08-02
> **Response to Reviewer zRmf**
>
> Thank you so much for your valuable feedback! Please see our response below.
>
> **Q1**: The evaluation seems limited - they only consider linear features. Perhaps this paper could benefit a little from a limitations section.
>
> **A1**: Thank you for the suggestion! We will add discussion on the limitations in the revision.
>
> Nevertheless, we study the linear case for simplicity of presentation and our result can indeed be extended to more general cases. For example, we can consider the case where the utility function is from a neural network (NN) class. The corresponding analysis and proof need to be modified in the following way:
> 1. We first need to use a class of Neural Tangent Kernels (NTK) to approximate the NN class. This is because our UCB-type algorithm requires the construction of bonus terms. Using NTK to approximate the NN-type utility functions allows us to construct these bonus terms. In terms of the regret bound, doing so will add an extra approximation error from approximating NN using NTK, which decays to zero as the number of neurons goes to infinity. See, e.g., [1].
> 2. We need to replace the logarithmic covering number of the linear function class, which is $d$,  by the logarithmic covering number of the NN class.
> 3. The dimension $d$ of the linear features will be replaced by the effective dimension of the NTK class. This dimension can be computed from the eigenvalues of the kernel.
>
> Both the logarithmic covering number and the effective dimension can be computed using the eigenvalue decay conditions of the NTK, which depends on the activation function and the distribution of the initial network weights.
> Therefore, our result can indeed be extended to more complicated and richer function classes.
>
>
> **Reference**:
> [1] Fine-grained analysis of optimization and generalization for overparameterized two-layer neural networks. Sanjeev Arora, Simon S. Du, Wei Hu, Zhiyuan Li, Ruosong Wang. ICML 2019.

---

> > ### Comment · Reviewer_zRmf · 2022-08-08
> > **Thanks!**
> >
> > Thanks so much for the response!

---

### Meta-Review · Area_Chair_RPzU · 2022-08-26

**Recommendation:** Accept
**Confidence:** Certain

**Metareview:**

This paper extends work on learning equilibria in matching markets from bandit feedback to more general Markov structure.  The paper brings to RL/MDP approaches to bear in this context.  An algorithm is presented and sublinear regret bound established.  This seems like a solid contribution and the reviewers are supportive of acceptance.

**Award:**

No

---

### Decision · Program_Chairs · 2022-09-14

Accept